# Non-pharmacological delirium detection and management interventions for informal caregivers of older people at home: A scoping review protocol

**Mary T. Fox**[1,2]*, **Ilo-Katryn Maimets**[3], **Jeffrey I. Butler**[1,2], **Souraya Sidani**[1], **Christina Godfrey**[4]

**1** School of Nursing, York University, Toronto, Ontario, Canada, **2** York University Centre for Aging Research and Education, Toronto, Ontario, Canada, **3** Steacie Science and Engineering Library, York University, Toronto, Ontario, Canada, **4** Queen's Collaboration for Health Care Quality: A JBI Centre of Excellence, School of Nursing, Queen's University, Kingston, Ontario, Canada

* maryfox@yorku.ca

**Data Availability Statement:** No datasets were generated or analysed during the current study. All

## Abstract

### Objective

The objective of this proposed scoping review is to identify and map the available evidence on interventions that aim to help informal caregivers identify and/or manage delirium in an older person at home.

### Introduction

Delirium is a neurocognitive condition characterized by acute confusion and is associated with increased risk of morbidity and mortality. Research estimates delirium to be present in 17% of community-dwellers aged 85 and older, increasing proportionally with age to 45% in those aged 90 and older. Delirium often occurs at the onset of an older person's acute illness or exacerbation of a chronic illness (sometimes while at home) and, because of its protracted nature, usually continues after a hospital stay. Even when an older person's delirium resolves during hospitalization, they remain at risk of its recurrence after discharge home. Consequently, knowing how to detect and manage delirium is critical for informal caregivers of older people at home. However, there are no reviews focused exclusively on this topic in this setting.

### Inclusion criteria

The population of interest includes informal caregivers of a person aged 65+. Concepts of interest include delirium detection and/or management interventions. The context of interest is any setting where informal care is delivered, including the transition from hospital to home, in any geographical area.

relevant data from this study will be made available upon study completion.

**Funding:** This scoping review is being supported by a York University Research Support Grant and Library Minor Research Grant, as well as the Research At York (RAY) program. NA Dr. Mary T. Fox and Ilo-Katryn Maimets.

**Competing interests:** The authors have declared that no competing interests exist. This scoping review is being supported by a York University Research Support Grant and Library Minor Research Grant, as well as the Research At York (RAY) program.

## Materials and methods

The review will be conducted according to the JBI guidelines for scoping reviews. A three-step search strategy will be used to locate both published and unpublished papers in MEDLINE, Embase, CINAHL, PsycINFO, Web of Science Core Collection, ProQuest Nursing & Allied Health, SCOPUS, LILACS, and SciELO, PQD&T, NDLTD, Google Scholar and Google. No language restrictions will be placed on the review. Papers will be screened for eligibility at the title, abstract, and full text level by two independent reviewers. Data will be extracted by two independent reviewers and managed in Covidence. Any disagreements in screening or data extraction will be resolved by consensus or a third reviewer. Results will be summarised in narrative and tabular formats.

## Introduction

Delirium is a serious neurocognitive syndrome characterized by a sudden and fluctuating disturbance in an individual's cognition, attention, and awareness [1]. Its diagnosis "requires evidence of an underlying organic cause" [2] (p. 1) which can involve any stressor (e.g., metabolic imbalance, infection) that disrupts the homeostasis of an individual with already diminished physiological reserves [1,2]. Although delirium can present at any age during the onset of an 1) acute illness, 2) acute exacerbation of a chronic condition, or 3) injury, it typically occurs in older people (age 65+) [3].

Although few studies have examined the prevalence of delirium at home in the community, studies estimate it ranges from .5% to 34.5% in the general non-hospital population aged 65 + with higher prevalence rates with increasing age [4]. For example, one study found delirium to be present in 17% of community-dwellers aged 85 and older increasing proportionally with age to 39% in those aged 95 and older [5]. Consequently, informal caregivers (i.e., any family member, friend, neighbor, or volunteer who provides informal, unpaid care at home; hereafter, referred to as caregivers) need to be able to recognize its onset at home in the community. Doing so will allow caregivers to seek medical attention from a primary care provider, which may prevent hospital admission and the worsening of delirium.

In the hospital setting, delirium has been found in 10%-30% of older patients upon admission to the emergency department [1], 20 to 23% of older medical inpatients [6,7], up to 50% of older post-operative patients, and 80% of older intensive care patients [8]. Delirium is regarded as reversible when the underlying cause can be resolved [3]. However, because of delirium's protracted nature, when it is diagnosed in hospital, it usually continues after discharge– 36% of older patients have delirium at the time of hospital discharge [9]. The probability of full recovery is only 4% at 4 weeks and 27% at 24 weeks post-discharge [10]. Consequently, the caregivers of patients discharged with delirium require information on non-pharmacological interventions, such as promoting physical activity and reducing environmental noise, which are needed to promote recovery at home following hospitalization. Yet, many caregivers recognize their lack of knowledge in managing delirium and they find caring for someone with delirium to be very challenging [11].

Even when an older person's delirium resolves during hospitalization, they remain at risk of experiencing a recurrence after discharge [12]. Because delirium can manifest as decreased psychomotor activity (e.g., listlessness, sleepiness), increased psychomotor activity (e.g., restlessness, agitation) or mixed psychomotor activity (e.g., alternating between listlessness and

restlessness) [13], it can be difficult for caregivers to identify and/or manage. For example, when an older person is lethargic and listless, a caregiver may fail to recognize delirium and not seek medical help, thereby putting the older person at risk of experiencing delirium-related complications. Delirium in the post-discharge period is associated with serious complications including functional disability, loss of independence [8], hospital readmission, and death [10]. Indeed, people who manifest their delirium through decreased psychomotor activity have worse outcomes than those who manifest their delirium through increased psychomotor activity–likely due to delayed detection [14]. Caregivers may interpret excessive sleepiness as a person's need for sleep and inadvertently encourage bed rest. Because ambulation and regulation of the sleep-wake cycle are interventions that promote recovery of delirium and mitigate its negative effects [3], encouraging bed rest may precipitate a new delirium or worsen an existing delirium.

While prior reviews have established the effectiveness of delirium detection [15–17] and/or management [18–20] interventions in improving patient outcomes, most reviews were limited to the hospital setting and did not extend into patients' homes during the post-discharge period [16–20]. We found only four reviews that included studies conducted with the caregivers of older adults at home. A review by Zhou and colleagues (2023) focused solely on the diagnostic accuracy of tools to detect delirium and did not describe if and how caregivers were taught to use the tool [21], and a 2015 review led by Carbone excluded studies that focused only on the detection of delirium [22]. A review by Lee's team (2023) described educational interventions for managing delirium but provided a high level, thematic interpretation of the interventions and did not describe their characteristics [23]. Only one review (Bull et al, 2016) attempted to describe the intervention under examination but only addressed dose (e.g., duration, frequency) [15]; moreover, that review was limited to studies examining the effects of educational interventions with experimental, quasi-experimental, and comparative designs. Most studies with these types of designs, particularly in the testing of non-pharmacological interventions, do not adequately describe the interventions being tested [24,25]. It has been our experience that detailed descriptions of interventions, when they do appear, tend to be published in protocols and other descriptive papers [26]. The lack of description of interventions in this clinical area thus likely limits intervention implementation and uptake [27]. For example, knowing whether an intervention initiated in hospital was in fact continued into the post-discharge period is critical to optimizing its use in the community.

Furthermore, the four available reviews restricted inclusion to studies published in English [15,21–23] or Chinese [21]. Delirium, however, is an important global issue as many countries are experiencing population aging [28]. There is likely untapped knowledge published in other languages that could enhance understanding of how to help caregivers detect and manage delirium. Yet, to the best of our knowledge, this information has not yet been synthesized to facilitate its widespread adoption.

Lastly, the four reviews that we found lacked comprehensiveness in their selection of databases. Bramer et al, 2017 recommends that, at a minimum, Medline and Embase should be searched along with Web of Science Core Collection, Google Scholar and specialized databases that are relevant to the topic [29]. In the case of searching for delirium detection and management interventions, CINAHL and PsycINFO should be included [29]. Likewise, any search in these databases should include a combination of index terms and keywords; only two reviews indicated that index terms and keywords were used to develop the search strategy [15,21].

This proposed scoping review will differ from prior reviews in several ways. First, the review will not be limited to only the hospital setting and will include interventions initiated in the hospital and extended into the home. Second, this review will also include studies that initiated the interventions in the non-hospital setting. Third, unlike prior reviews, this review will provide comprehensive, in-depth descriptions of the interventions. Fourth, the search strategy

will undergo peer review and will be as comprehensive as possible. Lastly, there will be no limitations on the language of papers.

The proposed review addresses an immense gap in synthesized knowledge. In our preliminary search of PROSPERO, MEDLINE, Pilot and Feasibility Studies, the Cochrane Database of Systematic Reviews, Epistemonikos, and the *JBI Evidence Synthesis*, we identified no current or underway systematic or scoping reviews on this specific topic. Furthermore, this search identified four unique articles that were not included in the four prior reviews that included studies conducted with the caregivers of older adults at home [15,21–23]. The objective of this proposed scoping review is to identify and map the available evidence on interventions that aim to help caregivers identify and/or manage delirium in an older person at home.

## Review questions

1. What non-pharmacological interventions that aim to help caregivers detect and/or manage delirium of an older non-institutionalized person at home have been studied?

    a. What outcomes have been examined (e.g., improvement in caregivers' knowledge of delirium, ability to recognize delirium, ability to manage delirium, burden, and stress)?

    b. How were outcomes measured?

    c. What results were reported?

2. What were the characteristics of the interventions that have been studied?

    a. What were the goals of the interventions (e.g., detect delirium, know what to do if delirium is detected)?

    b. What were the components and activities of the interventions (e.g., educational with teachback, behavioral with demonstration of a skill)?

    c. What materials were used in the interventions (e.g., educational pamphlets, videos)?

    d. What were the doses of the interventions (e.g., length, duration, frequency of intervention sessions)?

3. What was the context of implementing those interventions?

    a. In what setting were the interventions initiated (e.g., hospital, home, primary care clinic)?

    b. Who provided the interventions to caregivers (e.g., nurses, social workers)?

    c. What modes of providing the interventions to caregivers have been used (e.g., in-person, online)?

4. What were the characteristics of the caregivers that were included in the studies?

    a. What were the living arrangements of the caregivers (e.g., living with the patient)?

    b. What was the level of literacy of the caregivers (e.g., health and digital literacy, reading level)?

## Materials and methods

The proposed scoping review will be conducted in accordance with the JBI methodology for scoping reviews [30] and the Preferred Reporting Items for Systematic reviews and Meta-

Analyses extension for Scoping Reviews (PRISMA-ScR) [31]. A scoping review is "a type of evidence synthesis that aims to systematically identify and map the breadth of evidence available on a particular topic, field, concept, or issue, often irrespective of source (i.e., primary research, reviews, non-empirical evidence) within or across particular contexts" (page 1) [32]. Because we could not locate any existing reviews that provide an in-depth description of interventions that aim to help caregivers identify and/or manage delirium in an older person at home, we determined that a scoping review to identify and map this body of literature, rather than a systematic review, was most appropriate to meet our review objective. The protocol is registered with AsPredicted, Registration #160678 (https://aspredicted.org/WNW_QFS).

## Inclusion criteria

**Participants.** This review will include papers that targeted informal, unpaid caregivers who are family members, friends, volunteers, or neighbours providing care or support to an older person (aged 65+) at home. Providing care or support is defined as any type of assistance in basic or instrumental activities of daily living of any duration or frequency and need not be provided by live-in caregivers. Caregivers may be providing care and support to older people who:

1. received hospital care for various medical conditions and/or injuries and developed delirium and was discharged home;

2. received hospital care for various medical conditions and/or injuries and was at risk of developing delirium in the post-discharge period; and/or

3. are living at home and are at risk of developing delirium.

Papers will be excluded if informal caregiver data cannot be separated from those of patients and/or healthcare professionals.

**Concept.** The review will consider non-pharmacological interventions that aim to help caregivers detect and/or manage delirium. Interventions that exclusively aimed to help caregivers prevent delirium will be excluded.

**Context.** The context for the review is any setting where informal care is delivered, however, we will consider papers examining interventions that were initiated in the hospital (including the emergency department) provided they were continued at home after hospital discharge. All geographical areas will be eligible for inclusion. Papers written since 1980, the year that delirium became a diagnostic entity in the DSM-III, will be considered for inclusion [33]. Papers that focused exclusively on institutional settings such as nursing homes or settings staffed with full-time formal healthcare workers will not be eligible.

**Types of sources.** There will be two types of sources of evidence in this review: 1) papers reporting on the effectiveness of interventions that aim to help caregivers detect and/or manage delirium, and 2) papers describing those interventions. For the first type of source, we will consider studies that evaluated interventions, using any research design (e.g., randomized and non-randomized trials). For the second type of source, we will consider qualitative study designs, including qualitative descriptive, phenomenology, action research, grounded theory, ethnography, and mixed-method study designs. We will also include protocols, commentaries, abstracts, editorials, text, opinion, conceptual, clinical, and empirical papers describing the interventions; eligibility will be confirmed when the paper cites an intervention reported in an effectiveness study or the included effectiveness study cites the paper as providing additional information about the intervention, or when the study author (s) confirm that the paper describes the intervention. Papers focused exclusively on

delirium prevention interventions or psychometric testing of delirium detection tools will be excluded, as will diagnostic studies.

**Search strategy.**    The search strategy will be conducted according to the Preferred Reporting Items for Systematic reviews and Meta-Analyses literature search extension (PRISMA-S) [34]. A three-step search strategy will be used to locate both published and unpublished papers. Step 1: One of us (IM), an academic librarian with expertise in teaching and conducting reviews, will conduct an initial scoping search of MEDLINE (OVID) using only MeSH terms where feasible, to locate a set of relevant papers which will then be searched in PubMed for the purpose of harvesting additional MeSH terms and keywords from relevant results and gathering additional relevant papers from the "Similar articles" and "Cited by" lists. MeSH terms will also be searched in CINAHL to locate additional CINAHL-specific index terms and additional keywords from relevant articles. The final vocabulary selection will be done in consultation with team members with clinical expertise on the topic (see S1 Appendix). The final search strategy, including all identified keywords and index terms, will be submitted for peer-review to colleagues who, while not blinded, nevertheless will adhere closely to the Peer Review of Electronic Search Strategies (PRESS) guidelines for reviews [35]. Step 2: The search strategy will then be translated into the syntax of additional included databases, preserving strict fidelity to the original strategy so that index terms are translated individually to find the counterparts in the target databases, and the keyword strategy will be transferred as unchanged as possible taking database constraints and idiosyncrasies into account. Given the limitations of some of the grey literature databases and search engines, simplified keyword strategies (S1 Appendix) will be used to search various online sources such as Networked Digital Library of Theses and Dissertations (NDLTD) and Google that do not support more sophisticated search techniques. Advanced Search features of search engines and databases will be used where available. Step 3: Once articles, reports, and other grey literature information sources have been screened and identified for inclusion at the full text stage, their reference lists will be examined for any additional relevant papers that may meet the inclusion criteria for the review. All search strategies, as they are run in each database, will be reported in the scoping review.

To cast a broad net and capture articles from other cultures and jurisdictions, the databases to be searched will include MEDLINE (OVID), Embase (OVID), CINAHL (EBSCOhost), PsycINFO (OVID), Web of Science Collection, ProQuest Nursing & Allied Health, SCOPUS, LILACS, and SciELO and Google Scholar. A search for unpublished papers will be conducted in ProQuest Dissertation and Theses (ProQuest), NDLTD, and Google. Google search algorithms are not transparent and adapt themselves to the user; to avoid this adaptation by Google, the searchers will log off their Google account prior to searching [36]. The first 100 entries of the Google searches will be scanned for relevant results. No language restrictions will be placed on the review. Papers not published in English will be translated using DeepL.

**Source of evidence selection.**    Following the search, all identified citations will be collated and uploaded into Covidence and duplicates removed. After training and pilot testing of the screening process, titles and abstracts will be screened independently by two or more reviewers for assessment against the inclusion criteria for the review. The full text of selected citations will be retrieved and assessed in detail against the inclusion criteria by two or more independent reviewers. Reasons for exclusion of sources of evidence at full text will be recorded and reported in the scoping review. Any disagreements between the reviewers at any stage of the screening process will be resolved through discussion, or with an additional reviewer. The results of the search and the review process will be reported in full in the final scoping review and presented in a PRISMA flow diagram [37]. Papers not published in English will be translated using DeepL.

## Data extraction

Data will be extracted from papers included in the scoping review by using a data extraction tool developed by the reviewers, based on the Template for Intervention Description and Replication (TIDieR) [27] (S2 Appendix). Teams of two independent reviewers will pilot test the tool for ease of use, redundancy, clarity, and comprehensiveness of data extracted, and time for completion; we will perform the pilot test on two to three items for each type of evidence source (i.e., papers reporting on the effectiveness of the interventions and papers describing those interventions) [38]. Following the pilot test, we will modify the tool accordingly. Any modifications, and their rationale, made at this stage or throughout the scoping review (e.g., data items not initially considered but identified and determined by our team as relevant) [38] will be reported in the scoping review publication. Extracted data will include specific details about the participants, concept, context, study methods, and key findings relevant to the review questions and sub-questions. Any disagreements on the data extracted will be resolved through discussion or with an additional reviewer. We will contact authors of the papers to request any missing or additional data, where required.

## Data analysis and presentation

Data analysis will involve descriptive statistics including frequencies, counts, and measures of central tendency and variation (e.g., medians and ranges for duration of the intervention). Textual data will be subjected to conventional content analysis. Data presentation is planned to be in tabular format. A narrative summary will accompany tabulated and/or charted results and will describe how the results relate to the review objective, questions, and sub-questions. Covidence (Veritas Health Innovation) will be used to manage the references, abstracts, and full-text articles included in the scoping review.

## Supporting information

**S1 Appendix. Search strategy.**
(DOCX)

**S2 Appendix. Draft data extraction instrument.**
(DOCX)

**S3 Appendix. PRISMA-P 2015 checklist.**
(DOCX)

## Acknowledgments

The authors would like to thank York University undergraduate students Shannon Gordon and Madison Clancy for their help in reviewing the literature.

## Author Contributions

**Conceptualization:** Mary T. Fox.

**Data curation:** Ilo-Katryn Maimets.

**Formal analysis:** Jeffrey I. Butler.

**Methodology:** Mary T. Fox, Souraya Sidani, Christina Godfrey.

**Project administration:** Jeffrey I. Butler.

**Software:** Jeffrey I. Butler.

**Supervision:** Mary T. Fox.

**Writing – original draft:** Mary T. Fox.

**Writing – review & editing:** Mary T. Fox, Ilo-Katryn Maimets, Jeffrey I. Butler, Souraya Sidani, Christina Godfrey.

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
