## [Decision Letter · Decision Letter 0]

23 Jul 2024

PONE-D-24-24285Non-pharmacological delirium detection and management interventions for informal caregivers of older people at home: a scoping review protocolPLOS ONE

Dear Dr. Fox,

Thank you for submitting your manuscript to PLOS ONE. After careful consideration, we feel that it has merit but does not fully meet PLOS ONE’s publication criteria as it currently stands. Therefore, we invite you to submit a revised version of the manuscript that addresses the points raised during the review process.

The area of the proposed systematic review is of great interest, but the study protocol needs substantial revision prior to being considered for publication. I hope the reviewer's suggestions will help you in this process.

We look forward to receiving your revised manuscript.

Kind regards,

Pedro Kallas Curiati, M.D., Ph.D.

Academic Editor

PLOS ONE

Journal Requirements:

Reviewers' comments:

Reviewer's Responses to Questions

**Comments to the Author**

1. Does the manuscript provide a valid rationale for the proposed study, with clearly identified and justified research questions?

Reviewer #1: Yes

Reviewer #2: Yes

2. Is the protocol technically sound and planned in a manner that will lead to a meaningful outcome and allow testing the stated hypotheses?

Reviewer #1: Yes

Reviewer #2: Yes

3. Is the methodology feasible and described in sufficient detail to allow the work to be replicable?

Reviewer #1: Yes

Reviewer #2: Yes

4. Have the authors described where all data underlying the findings will be made available when the study is complete?

Reviewer #1: Yes

Reviewer #2: No

5. Is the manuscript presented in an intelligible fashion and written in standard English?

Reviewer #1: Yes

Reviewer #2: Yes

6. Review Comments to the Author

You may also provide optional suggestions and comments to authors that they might find helpful in planning their study.

Reviewer #1: This proposal addresses a highly relevant topic, particularly due to the universally high incidence of delirium. The methodology appears to be suitable and interesting; however, the authors' assertion that “no reviews have identified and mapped the available evidence on interventions that aim to help informal caregivers identify and/or manage delirium in an older person at home.” requires further clarification. For illustrative examples, please refer to the attached articles.

Young, J., Inouye, S. K., & Marcantonio, E. R. (2015). Delirium in older people. BMJ (Clinical research ed.), 350, h279.

We searched Ovid MEDLINE, Embase, and the Cochrane Library for the past 6 years, from January 1, 2011 until March 16, 2017 using a combination of controlled vocabulary and keyword terms. Since delirium is more prevalent in older adults, the focus was on studies in elderly populations; studies based solely in the intensive care unit (ICU) and non-English-language articles were excluded.

Advances in diagnosis can improve recognition and risk stratification of delirium. Prevention of delirium using nonpharmacologic approaches is documented to be effective, while pharmacologic prevention and treatment of delirium remains controversial.

Patnode CD, Perdue LA, Rossom RC, Rushkin MC, Redmond N, Thomas RG, Lin JS. Screening for Cognitive Impairment in Older Adults: An Evidence Update for the U.S. Preventive Services Task Force [Internet]. Rockville (MD): Agency for Healthcare Research and Quality (US); 2020 Feb. Report No.: 19-05257-EF-1. PMID: 32129963.

Caregiver and caregiver-patient dyad interventions including psychoeducation for the caregiver and care and case management interventions, reported in 88 trials (n=14,880), resulted in a consistent benefit on caregiver burden and depression outcomes. Effect sizes were mostly small, however, and were of unclear clinical significance. Little harm was evident in the few nonpharmacologic intervention trials that reported harms.

Reviewer #2: Review comments

Thank you for your protocol on a scoping review which will address an important, and under-researched topic. A few comments

1. Your introduction is written very much from the perspective of interventions that informal carers may continue at home, after a delirium in hospital. However, you also state that you will include papers for interventions commenced at home, where a person has developed delirium but not been admitted to hospital. I think the latter is potentially a very interesting area to explore, as interventions by informal carers may prevent hospital admission, and thus worsening of delirium. If you wish to include this scenario in your review, I suggest reframing your introduction.

2. In your introduction you write that there have been no reviews conducted on this topic but then say in an initial search you identified four studies which had not been included in previous reviews. Please clarify what these previous reviews were on, as it is not clear.

3. In outcomes (1) I would also include carers’ burden or stress, as this is likely to be a barrier to implementation of such interventions.

4. In outcomes (3) I would also include a question about what the characteristics of the caregivers were, if reported. i.e. is the intervention only feasible if the informal caregiver lives with the patient. Do they need a certain level of literacy / digital literacy.

5. Types of sources – this needs a little clarification and would read better if split up into individual statements, rather than use of multiple semicolons. You state that eligibility will be confirmed when the paper cites an intervention reported in an effectiveness study or the included effectiveness study cites the paper as providing additional information about the intervention, or when the study author(s) confirm that the paper describes the interventionyou will first search for any study which evaluates an intervention and then search for studies or grey literature which describes the intervention. Does the citation requirement refer just to the grey literature or also to qualitative studies as well? This will clearly narrow your inclusion criteria if so. Please then justify this in your methods.

6. Please describe where data will be available when the study is complete

7. PLOS authors have the option to publish the peer review history of their article (what does this mean?). If published, this will include your full peer review and any attached files.

Reviewer #1: No

Reviewer #2: **Yes: **Dr Anna Seeley

---

## [Author Response · Author response to Decision Letter 0]

30 Jul 2024

Journal Requirements

Comment: 1. Please ensure that your manuscript meets PLOS ONE's style requirements, including those for file naming. The PLOS ONE style templates can be found at

Response: We have carefully reviewed the manuscript to ensure it meets PLOS ONE's style requirements.

Comment: 2. We note that the grant information you provided in the ‘Funding Information’ and ‘Financial Disclosure’ sections do not match. When you resubmit, please ensure that you provide the correct grant numbers for the awards you received for your study in the ‘Funding Information’ section.

Response: We have ensured the ‘Funding Information’ and ‘Financial Disclosure’ sections match. The financial support provided did not include specific grant numbers.

Comment: 3. Please provide a complete Data Availability Statement in the submission form, ensuring you include all necessary access information or a reason for why you are unable to make your data freely accessible. If your research concerns only data provided within your submission, please write "All data are in the manuscript and/or supporting information files" as your Data Availability Statement.

Response: We have provided a complete Data Availability Statement in the submission form. Given that this is a protocol and does not report results, we selected 

“N/A” – No results are reported. We also indicate “No datasets were generated or analysed during the current study. All relevant data from this study will be made available upon study completion.”

Reviewer 1

Comment: This proposal addresses a highly relevant topic, particularly due to the universally high incidence of delirium. The methodology appears to be suitable and interesting; however, the authors' assertion that “no reviews have identified and mapped the available evidence on interventions that aim to help informal caregivers identify and/or manage delirium in an older person at home.” requires further clarification. For illustrative examples, please refer to the attached articles.

Young, J., Inouye, S. K., & Marcantonio, E. R. (2015). Delirium in older people. BMJ (Clinical research ed.), 350, h279.

We searched Ovid MEDLINE, Embase, and the Cochrane Library for the past 6 years, from January 1, 2011 until March 16, 2017 using a combination of controlled vocabulary and keyword terms. Since delirium is more prevalent in older adults, the focus was on studies in elderly populations; studies based solely in the intensive care unit (ICU) and non-English-language articles were excluded.

Advances in diagnosis can improve recognition and risk stratification of delirium. Prevention of delirium using nonpharmacologic approaches is documented to be effective, while pharmacologic prevention and treatment of delirium remains controversial.

Patnode CD, Perdue LA, Rossom RC, Rushkin MC, Redmond N, Thomas RG, Lin JS. Screening for Cognitive Impairment in Older Adults: An Evidence Update for the U.S. Preventive Services Task Force [Internet]. Rockville (MD): Agency for Healthcare Research and Quality (US); 2020 Feb. Report No.: 19-05257-EF-1. PMID: 32129963.

Caregiver and caregiver-patient dyad interventions including psychoeducation for the caregiver and care and case management interventions, reported in 88 trials (n=14,880), resulted in a consistent benefit on caregiver burden and depression outcomes. Effect sizes were mostly small, however, and were of unclear clinical significance. Little harm was evident in the few nonpharmacologic intervention trials that reported harms.

Response: We were unable to locate the recommended article by Young et al (2015). We believe that there may have been an error in the citation. However, we found the passage in a review article by Oh ES, Fong TG, Hshieh TT, Inouye SK. Delirium in older persons: advances in diagnosis and treatment. Jama. 2017 Sep 26;318(12):1161-74.The Oh et al, 2017 review article restricted studies to RCTs which would have precluded the retrieval of descriptive studies which our review attempts to overcome. Also, the review article restricted studies to those published in English in the 6-year period prior to 2017 – which are major limitations given that delirium is a global issue and there is likely untapped knowledge published in other languages that could enhance understanding of how to help caregivers detect and manage delirium. Lastly, this review article was not focused on studies aiming to help informal caregivers detect and manage delirium. Consequently, this lack of focus likely precluded finding this body of knowledge. 

Thank you for drawing our attention to the Patnode et al (2020) review. However, the review questions and findings pertained to dementia and mild cognitive impairment, not delirium. Here is a passage that provides evidence of this (page 26): “We identified 224 trials representing more than 50,000 patients and or caregivers and three cohort studies with more than 190,000 patients that address the treatment or management of MCI or mild to moderate dementia”.

Reviewer 2

Comment: Your introduction is written very much from the perspective of interventions that informal carers may continue at home, after a delirium in hospital. However, you also state that you will include papers for interventions commenced at home, where a person has developed delirium but not been admitted to hospital. I think the latter is potentially a very interesting area to explore, as interventions by informal carers may prevent hospital admission, and thus worsening of delirium. If you wish to include this scenario in your review, I suggest reframing your introduction.

Response: We have reframed the introduction by providing more information about the prevalence of delirium in the community-dwelling non-hospital population and its increasing prevalence with advancing age. We also explain that delirium often occurs at the onset of an older person’s acute illness or exacerbation of a chronic illness which may occur at home. We propose that caregivers need to be aware of the signs of delirium so that they can seek medical attention from their primary care providers which may prevent hospital admission and the worsening of delirium.

Comment: In your introduction you write that there have been no reviews conducted on this topic but then say in an initial search you identified four studies which had not been included in previous reviews. Please clarify what these previous reviews were on, as it is not clear.

Response: We now clarify that this search identified four unique articles that were not included in the four prior reviews that included studies conducted with the caregivers of older adults at home.

Comment: In outcomes (1) I would also include carers’ burden or stress, as this is likely to be a barrier to implementation of such interventions.

Response: We plan to synthesize the evidence on all caregiver outcomes including burden or stress. We have clarified this by including burden and stress as examples in the types of outcomes that may possibly be reported in studies.

a) What outcomes have been examined (e.g., improvement in caregivers’ knowledge of delirium, ability to recognize delirium, ability to manage delirium, burden, and stress)?

Comment: In outcomes (3) I would also include a question about what the characteristics of the caregivers were, if reported. i.e. is the intervention only feasible if the informal caregiver lives with the patient. Do they need a certain level of literacy / digital literacy.

Response: In the section on review questions, we now include: 

4. What were the characteristics of the caregivers that were included in the studies?

a) What were the living arrangements of the caregivers (e.g., living with the patient)?

b) What was the level of literacy of the caregivers (e.g., health and digital literacy, reading level)?

We opted not to locate this question in the outcomes to ensure conceptual clarity.

Comment: Types of sources – this needs a little clarification and would read better if split up into individual statements, rather than use of multiple semicolons. You state that eligibility will be confirmed when the paper cites an intervention reported in an effectiveness study or the included effectiveness study cites the paper as providing additional information about the intervention, or when the study author(s) confirm that the paper describes the intervention you will first search for any study which evaluates an intervention and then search for studies or grey literature which describes the intervention. Does the citation requirement refer just to the grey literature or also to qualitative studies as well? This will clearly narrow your inclusion criteria if so. Please then justify this in your methods.

Response: We have removed the semi-colon and have split the sentence up to enhance its readability. There is no citation requirement. We now number the criteria so that readers can clearly see that any one criterion can be met for the paper to be eligible. We now state “The eligibility of these descriptive papers will be confirmed when 1) the paper cites an intervention reported in an effectiveness study, 2) the included effectiveness study cites the paper as providing additional information about the intervention, or 3) the study author(s) confirm that the paper describes the intervention.

Comment: Please describe where data will be available when the study is complete

Response: We now explain that the data will be available from the first author upon request.

---

## [Editor Report · Decision Letter 1]

1 Aug 2024

Non-pharmacological delirium detection and management interventions for informal caregivers of older people at home: a scoping review protocol

PONE-D-24-24285R1

Dear Dr. Fox,

We’re pleased to inform you that your manuscript has been judged scientifically suitable for publication and will be formally accepted for publication once it meets all outstanding technical requirements.

Kind regards,

Pedro Kallas Curiati, M.D., Ph.D.

Academic Editor

PLOS ONE
---

## [Editor Report · Acceptance letter]

12 Sep 2024

PONE-D-24-24285R1 

PLOS ONE

Dear Dr. Fox, 

I'm pleased to inform you that your manuscript has been deemed suitable for publication in PLOS ONE. Congratulations! Your manuscript is now being handed over to our production team.

Kind regards, 

on behalf of

Dr. Pedro Kallas Curiati 

Academic Editor

PLOS ONE